# Pregnancy Risk, Infant Surveillance, and Measurement Alliance (PRISMA) Maternal and Newborn Health Study: protocol for a multisite, prospective, open cohort study of pregnancy and postpartum health outcomes in South Asia and sub-Saharan Africa

Pregnancy Risk, Infant Surveillance, and Measurement Alliance (PRISMA) Investigators

**Correspondence to**
Emily R Smith;
emilysmith@gwu.edu

## ABSTRACT

**Introduction** Maternal and child mortality has markedly decreased worldwide over the past few decades. Despite this success, the decline remains unequal across countries and is overall insufficient to meet the Sustainable Development Goals. South Asia and sub-Saharan Africa bear most of the burden of maternal and child morbidity and mortality. Major gaps persist in our understanding of the causes, timing, diagnostic thresholds and risk factors for adverse outcomes in these regions. Addressing these gaps requires new ways to prevent and treat disease, from novel diagnostics to precision public health strategies, all of which rely on high-quality clinical data from diverse populations. The Pregnancy Risk, Infant Surveillance, and Measurement Alliance (PRISMA) Maternal and Newborn Health Study aims to estimate population-level prevalence of morbidities and mortality and to assess biological, clinical and sociodemographic risk among mother–infant pairs in India, Pakistan, Kenya, Ghana and Zambia.

**Methods and analysis** This study is a prospective, open cohort study with a planned recruitment of about 6000 women annually across six research sites in five countries. Participants are pregnant women enrolled less than 20 weeks gestation, as determined by ultrasound, identified through active house-to-house and facility-based surveillance. Robust clinical data will be collected at 12 scheduled study visits during antenatal care, labour and delivery, and through 1 year postpartum. A total of 34 outcomes will be captured. The primary analysis will estimate the burden of adverse outcomes and examine associated risk factors to inform future intervention strategies. Data will also be used to develop normative values for pregnant and postpartum women, as well as predictive models to assess pregnancy risk.

**Ethics and dissemination** PRISMA received institutional and national ethical approvals. Findings will be published in peer-reviewed open-access journals and disseminated at national and international forums to inform clinical guidelines and public health practice.

---

### STRENGTHS AND LIMITATIONS OF THIS STUDY

⇒ Harmonised, multidimensional data collection on social, biological, clinical and environmental factors related to health and disease.

⇒ Large, community-based sample of an estimated 18 000 pregnant women and 15 300 live births over the first 3 years of the study.

⇒ Use of combined cutting-edge machine learning algorithms and predictive modelling approaches to better understand and predict risks.

⇒ Morbidity and mortality estimates may not be fully generalisable to other settings.

---

**Trial registration number** NCT05904145.

## INTRODUCTION

Globally, the maternal mortality rate has declined by more than 43% since 1990.[1] Despite this success, progress remains unequal between regions, with low-income and middle-income countries (LMICs) bearing excess burden.[2] Southern Asia and sub-Saharan Africa account for 86% of all maternal deaths and have some of the world's highest maternal mortality rates at 134 (80% uncertainty interval (UI) 118 to 155) and 545 (UI 477 to 654) per 100 000 live births, for each region, respectively.[3] The WHO estimates that between 2003 and 2009 the top causes of death in both regions were haemorrhage, hypertensive disorders and sepsis.[4] These three causes were responsible for over half of all maternal deaths. Indirect causes—such as HIV, malaria, tuberculosis, iron-deficiency anaemia or diabetes mellitus—accounted for

another 27.5% of maternal deaths.[4] However, due to challenges in event reporting and linking hospital records to national health surveillance systems, comprehensive data on the causes of maternal deaths and near-misses are lacking.[5–8]

Pregnancy-related and pre-existing conditions are a growing concern, not only given their potential to be life-threatening, but also due to the lasting effects they have on maternal and infant health and quality of life.[9 10] It is well evidenced that maternal morbidities put women at risk of future complications and impact an estimated 4% to 8% of all deliveries.[11] However, the true burden of maternal morbidities remains unclear, as these conditions frequently go undiagnosed or undocumented, most existing research focuses on a single sequela, and few large clinical cohorts have been conducted in LMICs. For example, a review of 59 studies (only four of which were conducted in Southeast Asia and sub-Saharan Africa) found that hypertensive disorders of pregnancy more than doubled the risk of ischaemic heart disease and heart failure later in life.[12] Similarly, gestational diabetes and maternal anaemia have been identified as risk factors for long-term cardiovascular diseases.[13 14] Another systematic review of 24 studies in high-income countries found that postpartum haemorrhage was linked to persistent physical, psychological and psychosocial problems, including mastitis, post-traumatic stress disorder, depression and delayed mother–infant bonding.[15] The narrow clinical and geographical focus of available research limits our understanding of the full range of risks women face during and after pregnancy, and how this varies across health settings. Multidimensional research that assesses biological, clinical, demographic and social determinants of health is likely to improve risk prediction, thereby preventing maternal morbidities and reducing their enduring repercussions.

Infant health and development also suffer the consequences of maternal morbidities.[10 16] Data from a WHO Global Survey of 172 461 singleton deliveries in 22 LMICs found that pre-eclampsia and pregestational diabetes significantly increased odds of provider-initiated and spontaneous preterm birth.[17] Pre-eclampsia and gestational diabetes have been connected to heightened risk of lifelong adverse outcomes among offspring, including increased susceptibility to hypertension, kidney disease, nutritional deficiencies, cardiometabolic events, obesity, attention deficit hyperactivity disorder and autism spectrum disorder.[18–21] Maternal infections and poor nutrition during pregnancy have been linked to suboptimal cognitive development for children.[22 23] To provide newborns with the best start to a healthy life, improvements in antenatal care are essential to appropriately manage pre-existing conditions and screen for risk factors. Ultimately, addressing maternal morbidities not only benefits women's health but further supports the long-term well-being of their children.

While early identification of morbidities and risk for adverse outcomes in the antenatal and postnatal periods is essential, correct diagnoses and risk prediction are hindered by the lack of empirical data underlying clinical guidance and risk stratification algorithms. Often these are predominantly derived from data collected among non-pregnant adults in limited geographies, most commonly the USA and China, due to concentrated investments in healthcare research and extensive electronic health systems.[24–27] Data disparities limit the generalisability and predictive accuracy when applied to peripartum women.[25] The rise of machine learning has further highlighted the issue, as generated predictive algorithms typically rely on datasets from high-income settings, restricting their applicability to LMICs.[24] High-quality, representative data are required to inform clinical guidance and risk prediction tools for maternal morbidities; better data are needed to inform normative ranges for perinatal depressive symptoms, blood pressure, biomarkers for hypertensive disorders of pregnancy and haemoglobin levels.[28–30] Addressing these evidence gaps by generating multidimensional global datasets for use in clinical benchmarking is critical to enabling accurate, equitable care for all women.

## Study rationale and objectives

Robust data on pregnancy risks—including medical history, clinical symptoms and diagnoses, behavioural and lifestyle factors, social and economic characteristics, environmental exposures and nutritional status—are essential to developing strategies to effectively manage pregnancy risks and improve outcomes in resource-constrained environments.[31 32] Population-based, multidimensional, longitudinal data of mother–infant pairs is severely lacking, particularly from LMICs. This hinders our ability to estimate the prevalence of morbidities and mortality, and to attribute their various risk factors among pregnant women and infants during the antenatal and postnatal periods.[31] We further lack the detailed, individual patient data needed to develop advanced diagnostic tools and risk prediction algorithms, as well as adequate baseline data needed to effectively monitor for adverse events in the context of introducing new devices, drugs and vaccines. Existing datasets have significant limitations in study designs or the completeness and accuracy of data collected. Often, data are self-reported, cross-sectional and most datasets are unable to be pooled in a harmonised way. These limitations make it difficult to achieve the robust datasets needed for analysis and for prioritising investments in research, development and policy or guideline changes.

We designed the Pregnancy Risk, Infant Surveillance, and Measurement Alliance (PRISMA) Maternal and Newborn Health Study in response to these critical gaps. At a high level, the study aims to evaluate pregnancy risk factors and their associations with adverse outcomes, including stillbirth, infant mortality and morbidity, and maternal mortality and morbidity. The resulting harmonised dataset will be used to estimate population-level disease and outcome burden in two South Asian and three

sub-Saharan African countries, which will be evaluated to identify any overlooked health concerns in each study area. Our data will inform subsequent research to understand priority issues and to test innovative strategies that optimise pregnancy outcomes, including defining more representative and gestational-age-specific reference values for select biomarkers. The primary objectives are as follows: (1) To provide population-based prevalence estimates of key maternal and infant health outcomes, which may inform the development of future diagnostics and interventions; (2) To improve the global understanding of key risk factors for morbidity and mortality among pregnant women, their fetuses and mother–infant pairs for up to 1 year postpartum and (3) To collect data to enable the application of novel analytical techniques to create risk prediction tools.

## METHODS AND ANALYSIS
### Study design
The PRISMA Maternal and Newborn Health Study is an international, multisite, prospective, open-cohort study. Leveraging population-based and facility-based pregnancy surveillance systems, we will identify, screen and enrol pregnant participants prior to 20 weeks gestation as confirmed by ultrasound. Enrolment began in September 2022 in Pakistan, November 2022 in Kenya, December 2022 in Ghana and Zambia, June 2023 in South India and December 2023 in North India. Detailed clinical data will be collected to characterise the health status of women and their fetuses and infants, including ultrasound imaging, laboratory testing, physical examination, validated health surveys, medical history by maternal recall and facility record abstraction. Data collection will continue during the antepartum, intrapartum and postpartum periods up to 1 year following delivery. Participants will be assessed at scheduled visits at <20, 20, 28, 32 and 36 weeks gestation, at labour and delivery, at 3 days postpartum, and 1, 4, 6, 26 and 52 weeks postpartum. Liveborn infants will be similarly assessed at birth, day three of life, and 1, 4, 6, 26 and 52 weeks old. Any additional visits or hospitalisations, for example, due to complications, will also be documented. In the event of a miscarriage or stillbirth, participants will be followed up through 6 weeks and 1 year, respectively. Verbal autopsies will be conducted in the case of maternal, fetal, neonatal or infant death and analysed by InterVA computer software to ensure a harmonised approach to cause of death determination. The study protocol will be implemented uniformly across sites to ensure that all laboratory and diagnostics tests, primary health outcomes and related risk factors are measured correctly and consistently.

### Study outcomes
We aim to capture data on 34 adverse health outcomes (table 1), which are defined in online supplemental file. Outcomes were selected based on high-priority research gaps within study contexts and determined via an interactive protocol development process with input from study investigators and other subject matter experts. Feasibility was also considered when selecting the measurement method. Clinical definitions, measurements and constructed variables for each health outcome are standardised across study sites.

**Table 1** Study outcomes by population and level

| Outcome level | Population | |
| --- | --- | --- |
| | Gravida/puerperium | Fetus/infant |
| Primary | 1. Maternal mortality<br>2. Composite severe maternal outcomes<br>3. Maternal anaemia | 4. Stillbirth<br>5. Neonatal mortality<br>6. Low birth weight<br>7. Preterm birth<br>8. Small-for-gestational-age |
| Secondary | 9. Late maternal mortality<br>10. Pre-eclampsia<br>11. Gestational hypertension<br>12. Postpartum hypertension<br>13. Medical indication for preterm delivery<br>14. Preterm premature rupture of membranes<br>15. Gestational diabetes<br>16. Maternal infection and sepsis<br>17. Perinatal depression | 18. Fetal death<br>19. Infant mortality<br>20. Cause of neonatal mortality<br>21. Timing of neonatal mortality<br>22. Cause of stillbirth<br>23. Timing of stillbirth<br>24. Hyperbilirubinaemia<br>25. Neonatal sepsis<br>26. Possible severe bacterial infection<br>27. Postnatal weight trajectory<br>28. Infant growth |
| Tertiary | 29. Placental disorders<br>30. Uterine rupture<br>31. Prolonged labour<br>32. Unplanned surgery<br>33. Antepartum and postpartum haemorrhage | 34. Perinatal birth asphyxia |

## Study settings

Data collection will occur in six study areas: Kisumu and Siaya, Kenya; Kintampo, Ghana; Lusaka, Zambia; Vellore, India; Hodal, India; and Rehri Goth and Ibrahim Hyderi, Karachi, Pakistan. Protocol implementation is jointly carried out by clinical teams at participating health facilities and study staff from partnering research institutions (ie, the Kenya Medical Research Institute, Kintampo Health Research Centre, University of North Carolina Global Projects Zambia, Christian Medical College Vellore, Society for Applied Studies and Aga Khan University). Clinical assessments and subsequent findings conducted by study staff will be integrated with clinical care. Researchers at the George Washington University are responsible for protocol harmonisation and central data coordination. Consortium institutions and investigators were appointed by the Gates Foundation based on existing research infrastructure and expertise. Each country team identified and mapped geographical areas in which to conduct the study. Estimates of the study area population size are described in table 2.

## Eligibility criteria

Women who meet the following criteria may be eligible for enrolment: lives within the study catchment area with no plans to permanently relocate during pregnancy or within 1 year postpartum; meets minimum age requirement for consenting in country of residence (Ghana: 15 years of age; Kenya: 18 years of age or emancipated minors; Pakistan: 15 years of age or emancipated minors; Zambia: 15 years of age; India: 18 years of age) and viable intrauterine pregnancy less than 20 weeks gestation determined by ultrasound. Women who have more than one pregnancy during the study period are eligible for re-enrolment. Multiple gestations (ie, two, three or more fetuses) will also be eligible.

## Pregnancy surveillance

Household-based and facility-based surveillance systems will be used to identify women of reproductive age with suspected pregnancies. Routine health and demographic surveillance systems, either existing or established for the purpose of this study, will be leveraged to visit homes in the study catchment area at least once every 6 months. Women identified at these visits will be invited to provide basic socio-demographic information as a part of the population-based surveillance and screened for study eligibility. Women who receive antenatal care at participating health facilities will also be identified and screened. Combined, these women make up the sampling frame for the study cohort.

## Participant identification and enrollment

A multiphase process of prescreening, consenting and screening will be used to enrol participants (figure 1). Identified women of reproductive age will be prescreened for potential eligibility by community fieldworkers based on the following criteria: reported symptoms or clinical signs of pregnancy; estimated gestational age per last menstrual period <25 weeks; lives within the study catchment area; and meets minimum age requirement in each study site. Any woman potentially eligible after prescreening will be invited to provide informed consent or assent in her preferred language. Certified sonographers will then conduct an ultrasound to verify that the woman has an intrauterine pregnancy and is prior to 20 weeks gestation. Women who are beyond the gestational age threshold or with a non-viable pregnancy (eg, ectopic) are not eligible to participate and will be referred.

## Study follow-up procedures

Follow-up is planned to occur through 1 year postpartum for participants with liveborn and stillborn infants, and through 6 weeks for participants with spontaneous or induced abortions. Data will be collected at 12 study visits scheduled according to gestational age. This includes five antenatal study visits at <20 weeks (enrolment), 20 weeks, 28 weeks, 32 weeks and 36 weeks gestation. Data will also be collected for labour and delivery within 24 hours for facility births and 72 hours for home births. Deliveries will be observed by study staff, when possible, otherwise data will be abstracted from medical records and by maternal or provider recall. After delivery, there will be six postnatal visits at 3 days, 1 week, 4 weeks, 6 weeks, 26 weeks and 52 weeks postpartum for both mothers and infants. For each scheduled study visit there is an accepted visit window. If

| Table 2 | Estimates of study area population size, pregnancies and live births | | | |
|---|---|---|---|---|
| Study area | No. households | No. women of reproductive age | Total population size | Anticipated no. pregnancies (live births)* |
| Kintampo, Ghana | 34 640 | 45 767 | 177 200 | 3000 (2250) |
| Vellore, India | 54 435 | 47 443 | 234 624 | 2700 (2300) |
| Hodal, India | 36 321 | 35 499 | 196 678 | 2000 (1700) |
| Kisumu, Kenya | 44 932 | 47 826 | 171 698 | 3000 (2550) |
| Karachi, Pakistan | 23 170 | 34 682 | 227 549 | 6000 (5100) |
| Lusaka, Zambia | 31 165 | 34 942 | 145 263 | 2250 (1914) |
| All sites | 224 663 | 246 159 | 1 153 012 | 18 550 (15 814) |

*Anticipated number of pregnancies based off expected annual enrolment over 3 years; expected 85% of pregnancies to result in a live birth.

     Pregnancy Risk, Infant Surveillance, and Measurement Alliance (PRISMA) Investigators. *BMJ Open* 2026;**16**:e104512. doi:10.1136/bmjopen-2025-104512

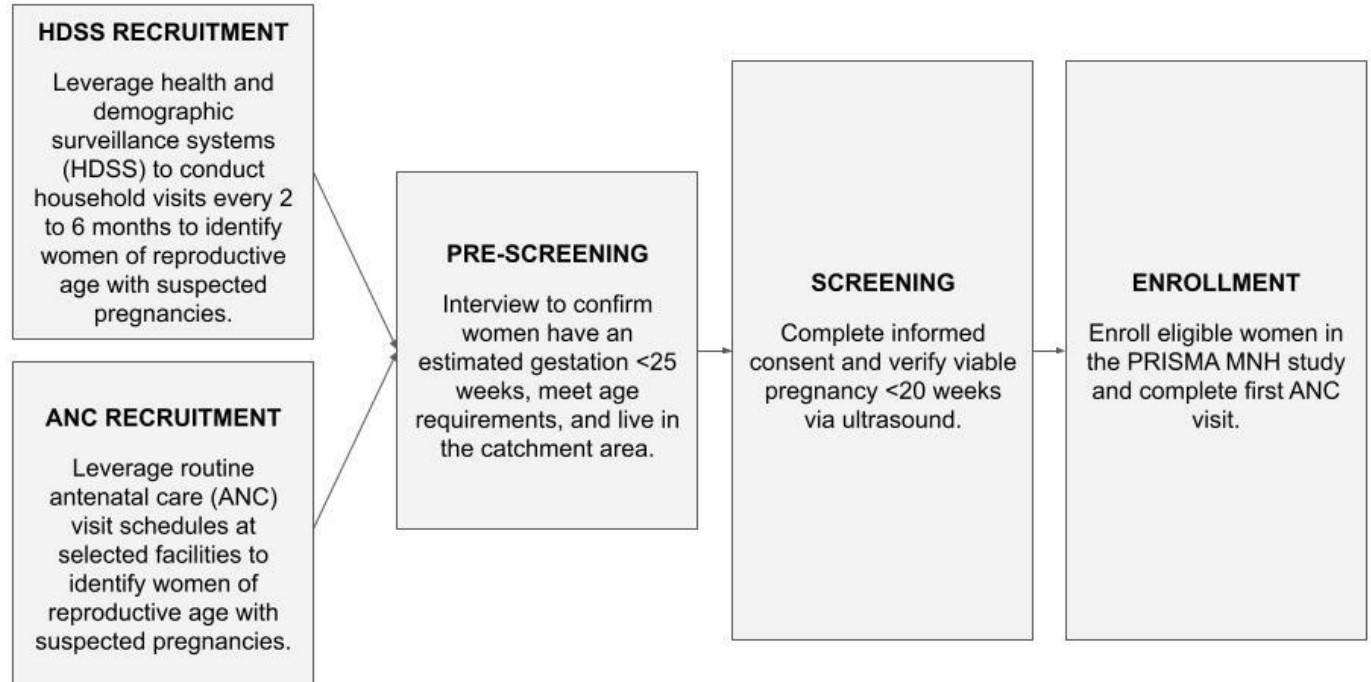

**Figure 1** Overview of surveillance and identification of potentially eligible women. MNH, Maternal and Newborn Health; PRISMA, Pregnancy Risk, Infant Surveillance, and Measurement Alliance.

the accepted window has passed and no visit occurred, the visit will be considered missed, forms submitted with any available information, and the subsequent visit scheduled. In addition to scheduled visits, any unscheduled follow-ups or hospital visits during the study period will be documented.

### Harmonized data collection

Trained study staff will use harmonised measurement procedures and data collection tools developed for the purpose of this study.[33] Each tool corresponds to a scheduled or ad hoc study visit conducted either at a study facility or at home and contains information on approximately 4000 core variables. For maternal participants, the scope of data collected during antenatal care includes sociodemographic characteristics, health behaviours, environmental exposures, medical and obstetric history, anthropometry, vital and clinical status, current physical and mental health state, medications and treatments, vaccination status, laboratory tests, ultrasounds, and diagnostic assessments (table 3).

As participants may deliver outside of the study facility, data capturing clinical status and events during labour and delivery will be collected via a combination of observation, retrospective self-report or report by birth attendant and/or abstraction from medical records. Most maternal assessments will be repeated during scheduled study visits in the postpartum period (table 4). For infant participants, data collected during postnatal care visits includes anthropometry, signs and symptoms of infection, clinical and vital signs, vaccination status, medications and treatments, and cognitive development assessments (table 4).

### Laboratory and diagnostic tests

Maternal and infant participants will undergo study-specific laboratory and diagnostic testing at scheduled visits (table 5). Wherever possible and with consideration to participant burden, gold standard methods will be used. Haemoglobin levels will be assessed at every antenatal and postnatal visit, and routine urinalysis will be done by dipstick at all antenatal visits. In addition to these assessments, each study contact point has specific requirements. Results will be provided in real time to participants for all ultrasound exams, rapid diagnostic tests and as available for blood tests. Where specified, specimens may be stored for later batch testing, in which case results will still be communicated with participants and subsequent referrals made as appropriate. Soluble fms-like tyrosine kinase-1 and placental growth factor biomarkers will not be used in clinical decision-making, as these are under investigation for clinical utility.

During the enrolment visit, pregnancy viability and fetal biometry will be determined via ultrasound for gestational age dating and to verify study eligibility. Blood tests will be done for haemoglobin A1c levels to assess for diabetes. Additional blood specimens will be collected to test for Rh factor, micronutrient deficiencies, inflammatory markers, liver and renal function, soluble fms-like tyrosine kinase-1 and placental growth factor. Rapid diagnostic tests will be conducted for HIV, malaria, syphilis, and hepatitis B and C infection. If the syphilis rapid test is positive, it will be followed up with a nontreponemal test to check syphilis titers. If clinically indicated by the WHO Four-Symptom Screen, a tuberculosis test will also be done.[34]

**Table 3** Type of data collected at scheduled antenatal study visits

| Type of data collected | Scheduled study timepoint* (✓ required; O optional) | | | | |
| --- | --- | --- | --- | --- | --- |
| | <20 weeks | 20 weeks | 28 weeks | 32 weeks | 36 weeks |
| Prescreening | ✓ | | | | |
| Ultrasound exam | ✓ | O | O | ✓ | O |
| Study enrolment | ✓ | | | | |
| Sociodemographics | ✓ | | | | |
| Clinical status | ✓ | ✓ | ✓ | ✓ | ✓ |
| Anthropometry | ✓ | ✓ | ✓ | ✓ | ✓ |
| Vital signs | ✓ | ✓ | ✓ | ✓ | ✓ |
| Lab tests | ✓ | ✓ | ✓ | ✓ | ✓ |
| Depression screen† | | ✓ | | ✓ | |
| Quality of life‡ | | | | ✓ | |
| Quality of care | | | | | O |

*Additional ad hoc forms are administered in the event of hospitalisation, severe adverse events, protocol deviations, and death (verbal autopsy).
†Depressive symptoms measured by an adapted version of the Edinburgh Postnatal Depression Scale.
‡Measured by the 12-Item Short Form Health Survey and WHO Disability Assessment Schedule.

At the 28-week antenatal visit, an oral glucose tolerance test will be done to screen for gestational diabetes among participants without previously diagnosed or overt diabetes (ie, normal haemoglobin A1c levels at enrolment). Blood specimens will be stored for soluble fms-like tyrosine kinase-1 and placental growth factor biomarker testing. A repeat ultrasound to measure fetal biometry is recommended at 28 weeks and required at 32 weeks. At the 32-week antenatal visit, blood tests will also be repeated for micronutrient deficiencies, inflammatory markers and thyroid function. If the syphilis rapid test is negative at enrolment, it will be repeated. If the rapid test is positive, it will be followed up with a nontreponemal test to check syphilis titers. At the 36-week antenatal visit, the rapid diagnostic tests for HIV, malaria, syphilis and tuberculosis infection are repeated as clinically indicated or locally recommended.

At the 6-week postnatal visit, an additional maternal blood specimen will be collected to test for micronutrient deficiencies and inflammation. For participants who tested positive for gestational diabetes during antenatal care, an oral glucose tolerance test will be done at the 6-week visit along with a haemoglobin A1c test at the final 52-week visit.

Further testing among a subsample or time-limited testing of specific infections or biomarkers will be incorporated at the discretion of the PRISMA consortium to better understand novel findings or previously undetected health concerns. Beginning in November 2023, sites asynchronously rolled out vaginal swab testing for chlamydia and gonorrhoea at the enrolment and 32-week antenatal care visits, which will continue for 1 year of data collection. Among a subsample of 1000 participants per site, testing for differential diagnosis of febrile illnesses—including hepatitis E, Zika, chikungunya, dengue and leptospirosis infection—will be administered at enrolment, 32 weeks gestation and 6 weeks postpartum. Depending on the identified burden of disease, investigators may choose to continue testing.

Infants similarly will undergo a limited set of laboratory and diagnostic tests at scheduled time points (table 6). At birth, transcutaneous bilirubin assessment is done within 72 hours (24 hours preferred). Transcutaneous bilirubin assessments will be repeated at the 3–5-day visit (<1 week) and the 1-week visit. Haemoglobin levels will be assessed by HemoCue at the 6-week, 6-month and 1-year visits. Tests for malaria infection will be performed at 6 months and 1 year. Infants will be tested for HIV and hepatitis B tests only in the case of maternal infection.

### Quality control and assurance

All procedures will adhere to Quality Assurance and Quality Control Standard Operating Procedures that were collaboratively developed for this study. Central training was conducted for designated study staff members from each site to ensure harmonised procedures for ultrasound examinations, verbal autopsies, and assessment for integrated management of childhood illnesses. Personnel taking anthropometric measurements received study-specific training and will participate in ongoing standardisation exercises at regular intervals. Calibration checks of circumference tapes, scales, stadiometers and infantometers for anthropometric assessments will be performed monthly. An external monitoring group has been contracted to ensure that the rights and safety of study subjects are protected; to ensure the quality and integrity

**Table 4** Type of data collected at scheduled delivery and postnatal study visits

| Type of data collected | Delivery | Postpartum <1 week | 1 week | 4 weeks | 6 weeks | 26 weeks | 52 weeks |
|---|---|---|---|---|---|---|---|
| **Maternal** | | | | | | | |
| Clinical status | ✓ | ✓ | ✓ | ✓ | ✓ | ✓ | ✓ |
| Anthropometry | ♦ | | | | ✓ | ✓ | ✓ |
| Vital signs | ✓ | ✓ | ✓ | ✓ | ✓ | ✓ | ✓ |
| Lab tests | O | ✓ | ✓ | ✓ | ✓ | ✓ | ✓ |
| Depression† | | | | | ✓ | | |
| Quality of life‡ | | | | ✓ | | | |
| Delivery outcome | ✓ | | | | | | |
| Quality of care | O | | | | | | O |
| Study close-out | | | | | ** | | ✓ |
| **Infant** | | | | | | | |
| Birth outcome | ✓ | | | | | | |
| Apgar score | ✓ | | | | | | |
| Anthropometry | ♦ | ✓ | ✓ | ✓ | ✓ | ✓ | ✓ |
| Jaundice screen§ | ♦ | ✓ | ✓ | | | | |
| Vital signs¶ | | ✓ | ✓ | ✓ | ✓ | ✓ | ✓ |
| Clinical status | | ✓ | ✓ | ✓ | ✓ | ✓ | ✓ |
| Lab tests | | | | | ✓ | ✓ | ✓ |
| Vaccination | | ✓ | ✓ | ✓ | ✓ | ✓ | ✓ |
| Cognitive development** | | | | | | | ✓ |
| Stimulation†† | | | | | | | O |
| Study close-out | | | | | | | ✓ |

Scheduled study time point* (✓ required; O optional; ♦ only if in study facility)

✓Required.
OOptional.
♦Only collected if delivery occurred in study facility.
**Close-out occurs at 6 weeks in the case of spontaneous or induced abortion.
*Additional ad hoc forms are administered in the event of hospitalisation, severe adverse events, protocol deviations and death (verbal autopsy).
†Depressive symptoms measured by an adapted version of the Edinburgh Postnatal Depression Scale.
‡Measured by the 12-Item Short Form Health Survey (SF-12) and WHO Disability Assessment Schedule.
§Measured by Mennen Medical Bili-Care transcutaneous bilirubin device and Bili-ruler visual icterometer.
¶Heart rate and oxygen saturation.
**Measured by the Global Scale for Early Development SF (version 1.0).
††Measured by the Family Care Indicators assessment.

of study data; and to ensure the study is conducted in compliance with the approved protocol and applicable regulatory requirements.

A proportion of ultrasound images will undergo external quality assessment, and all fetal biometry measures will undergo plausibility checks. Sonographers at all sites will be certified in study-specific procedures and undergo annual refresher training to ensure compliance. We will use Tricefy for secure cloud medical imaging storage and sharing of anonymised data. Anonymised images will be randomly peer-reviewed and expert-reviewed; summary quality reports will then be generated monthly.

Biospecimen collection and laboratory testing will be conducted according to Standard Operating Procedures. Laboratories implementing micronutrient measurements on-site will be enrolled in the Centers for Disease Control Performance Verification Program for Serum Micronutrients. For full blood count measurement, all participating laboratories will be enrolled in the UK National External Quality Assessment Scheme for Haematology. For full

**Table 5** Maternal laboratory and diagnostic tests during antenatal and postnatal care

| | Scheduled study visit | | | | | |
| | Antenatal care (weeks) | | | | Postnatal care (weeks) | |
| Assessment | <20 | 28 | 32 | 36 | 6 | 52 |
| --- | --- | --- | --- | --- | --- | --- |
| Haemoglobin levels* | ✓ | ✓ | ✓ | ✓ | ✓ | ✓ |
| Urinalysis* | ✓ | ✓ | ✓ | ✓ | | |
| Rh factor | ✓ | | | | | |
| Haemoglobin A1c (HbA1c) | ✓ | | | | | * |
| Oral glucose tolerance | | ✓† | | | * | |
| Micronutrient deficiencies‡ | ✓ | | ✓ | | ✓ | |
| Inflammation biomarkers§ | ✓ | | ✓ | | ✓ | |
| Preeclampsia biomarkers¶ | ✓ | ✓ | ✓ | | | |
| Liver function | ✓ | | | | | |
| Renal function | ✓ | | | | | |
| Thyroid function | | | ✓ | | | |
| Syphilis infection | ✓ | | ✓ | * | | |
| Syphilis titers | * | | * | | | |
| Chlamydia infection | + | | + | | | |
| Gonorrhoea infection | + | | + | | | |
| HIV infection | ✓ | | | * | | |
| Malaria infection | ✓ | | | * | | |
| Tuberculosis infection | * | | | * | | |
| Hepatitis B infection | ✓ | | | | | |
| Hepatitis C infection | ✓ | | | | | |
| Hepatitis E infection | + | | + | | + | |
| Zika infection | + | | + | | + | |
| Chikungunya infection | + | | + | | + | |
| Dengue infection | + | | + | | + | |
| Leptospirosis | + | | + | | + | |

✓Required.
*As clinically indicated.
+Testing completed for a subset of participants at each site, following which investigators will decide whether or not to continue with these tests based on the observed burden of disease.
*Select visits not shown due to limited labs required: antenatal visit at 20 weeks (haemoglobin and urinalysis only) and postnatal visits at <1, 1, 4, and 26 weeks (haemoglobin only).
†Only among women without previously diagnosed or overt diabetes (ie, normal HbA1c at enrolment).
‡Vitamin $B_{12}$ total cobalamin, folate, vitamin A retinol binding protein, iodine thyroglobulin, iron serum transferrin receptor and iron ferritin.
§C reactive protein and alpha 1-acid glycoprotein.
¶Soluble fms-like tyrosine kinase-1 and placental growth factor.

blood count and all other analytes, laboratories must also be enrolled in the College of American Pathologists' external quality assurance programme.

**Data quality monitoring**

To ensure the success of the study, a baseline phase was carried out in 2021 to pilot the performance of study tools and procedures. Performance was evaluated using both qualitative and quantitative methods. At the end of the pilot period, a harmonisation process to standardise tools and data systems across study sites was implemented.

Obtaining high-quality data is central to the study; thus, data quality monitoring is a key component of standard operating procedures. Each country team will be responsible for the initial data cleaning, including running of routine range and consistency checks as well as periodic reviews of distributions and identification of outliers. Any inconsistencies within their database will be resolved in consultation with the clinical and field data collection team. A cleaned, limited dataset from all sites will be stored in a secure central data repository cloud server,

**Table 6** Infant laboratory and diagnostic tests at birth and during postnatal care

| | Scheduled study visit | | | | | | |
|---|---|---|---|---|---|---|---|
| | Birth | Postnatal (weeks) | | | | | |
| Assessment | | <1 | 1 | 4 | 6 | 26 | 52 |
| Bilirubin levels* | ✓ | ✓ | ✓ | | | | |
| Haemoglobin levels | | | | | ✓ | ✓ | ✓ |
| Malaria infection | | | | | | ✓ | ✓ |
| HIV infection | | | | | * | * | * |
| Hepatitis B virus | | | | | | | * |

✓Required.
*In the case of maternal infection.
*Bilirubin measured via transcutaneous bilirubinometer and Bili-ruler visual icterometer.

shared at biweekly intervals. Data analysts on the coordination team will run additional data quality checks and produce both data queries and monitoring reports using pooled data on at least a bimonthly basis. Data queries will identify issues related to the following: range checks, skip patterns, consistency logic, date and time concordance, submitted records to the database, uniqueness of a record (ie, deduplicating), outliers, digit preference and missingness. Monitoring reports will include details such as rate of enrolment, protocol compliance, completeness of key clinical and lab assessments, and the distribution of key health outcomes. Data management teams will share codes for data checks via GitHub repositories.

### Data management systems and security
Data collection instruments were developed by each country's team. All systems are encrypted to ensure no subject information is readily accessible. Signed consent forms and study data collection forms will be stored in a secure location with access limited to authorised research staff. A limited study dataset, which will not contain any protected health information or personally identifiable information beyond complete birth and death dates, will be maintained in a secure central repository. This dataset may be shared with current and future research partners or affiliates for analysis purposes through a data-sharing agreement. Any partners given permission to access the data will be required to adhere to the Gates Foundation Global Access policies, which require that any knowledge and information gained or any products developed must be made available and accessible at an affordable price to the people most in need living in LMICs.

### Patient and public involvement
Patients and/or the public were involved to varying degrees at different study sites in the design of this research, including as participants on community advisory boards that were consulted. They were not directly involved in study recruitment. When selecting the primary and secondary outcomes, the prevalence and impact of each outcome within study site contexts was strongly considered. Additionally, the field data collection teams and study investigators considered the needs and feedback from community members at each step of the study protocol development process. For example, invasive clinical assessments were kept to a minimum to reduce unnecessary burden on participants. Local women were involved in the validation of multiple study tools, including the depression screening questionnaire, and shared their experiences of the assessments through cognitive debriefing interviews. On completion of the study, we intend to share the main findings with participants and community members via appropriate local dissemination methods. Finally, we will ensure that participants' contributions to this research are acknowledged in subsequent publications.

## ETHICS AND DISSEMINATION
### Ethical and safety considerations
This protocol, the informed consent documents and any subsequent modifications will be reviewed and approved by the relevant institutional review board (IRB) and ethics review committee (ERC) responsible for oversight of the study at each site. Approvals were received from the following IRBs and ERCs in each country: Ghana (Ghana Health Service ERC: 019/09/20; Kintampo Health Research Centre Institutional Ethics Committee: KHRCIEC/2020-17); India (Office of Research, Christian Medical College, Vellore, India: IRB14553; Ethics Review Committee, Society for Applied Studies, New Delhi: SAS/ERC/ReMAPPStudy/2022; Kenya (KEMRI Scientific and Ethics Review Unit: KEMRI/SERU/4166; Liverpool School of Tropical Medicine Ethics Committee: 23-020 Jaramogi Oginga Odinga Teaching and Referral Hospital (JOOTRH) Institutional Scientific Ethics Review Committee ISERC: ISERC/JOOTRH/549/2024); Pakistan (The Aga Khan University ERC: No. 2023-9100-26333; Pakistan National Institutes of Health—Health Research Institute, National Bioethics Committee: 4-87/NBCR-1023/23/973); Zambia (University of North Carolina Chapel Hill Office of Human Research Ethics: Study No. 14-2113; University of Zambia Biomedical

Research Ethics Committee: 016-04-14) and the USA (The George Washington University IRB: NCR224396; Columbia University IRB: IRB-AAAU7504; Harvard University IRB: IRB23-1093; The University of Alabama at Birmingham IRB: IRB-300013081). All pregnant women will be provided with information about the study and the option to opt out of data collection. For women who are not of legal age for consent, assent together with parental or other legal consent must be obtained per local standards. Study participants will be referred to higher-level or specialty care according to site-specific standard operating procedures.

## Dissemination plan

The data generated in the course of the study will be reviewed on an ongoing basis, according to priorities set by the investigators. Quarterly and annual reports will be shared with key stakeholders, as well as the funder (Gates Foundation). Interim findings with potential implications for informing public health policy or practice will be shared with major local and international stakeholders, including the WHO, US Centers for Disease Control, US Agency for International Development and Ministries of Health in each participating country. Local meetings will be organised to share the findings of this study with community members and study participants at each site. Abstracts from both country-specific and pooled global data analyses will be developed for dissemination through both local and international scientific conferences and publication in peer-reviewed journals.

**Acknowledgements** The Maternal and Newborn Health Study is a research project of the Pregnancy Risk, Infant Surveillance, and Measurement Alliance (PRISMA) Consortium. PRISMA members are grateful to the extensive network of individuals and organisations who contributed to the design of this study, including but not limited to biostatisticians, laboratory technicians, obstetricians, community health workers, field data collectors and research partners at the Institute for Health Metrics and Evaluation, Centers for Disease Control and Prevention, and the WHO. The authors also acknowledge the efforts of study participants and community members, without whom this would not be possible.

**Collaborators** The Pregnancy Risk, Infant Surveillance, and Measurement Alliance (PRISMA) Investigators include: Emily R. Smith, Christopher N. Mores, Qing Pan, Jennifer Seager, Sasha G. Baumann, Jaime Marquis, Christopher R. Sudfeld, Zahra Hoodbhoy, Muhammad Imran Nisar, Fyezah Jehan, Aneeta Hotwani, Nida Yazdani, Amna Khan, Farzana Shaheen, Kinza Farooqui, Leena Chatterjee, Arjun Dang, Manavi Dang, R Venketeshwar, Sarmila Mazumder, Neeraj Sharma, Arun Singh Jadaun, Rupa Talukdar, Blair J. Wylie, Victor Akelo, Winnie K. Mwebia, Joyce Were, Dickson Gethi, Zacchaeus Abaja, Gregory Ouma, Harun Owuor, Kephas Otieno, Anne George Cherian, Santosh Joseph Benjamin, Venkata Raghava Mohan, Balakrishnan Vijayalekshmi, Sunitha Varghese, Jasmine Sugirtha, Daniel Jebakumar, James A, Margaret P. Kasaro, M. Bridget Spelke, Wilbroad Mutale, Felistas Mbewe, Humphrey Mwape, Bethany Freeman, Bellington Vwalika, Mutale Sampa, Kwaku Poku Asante, Sam Newton, Charlotte Tawiah Agyemang, Irene Apewe Adjei, Veronica Agyemang, Dennis Adu-Gyasi, Eliezer Odei-Lartey, Stephaney Gyaase, and Laura M. Lamberti.

**Contributors** This study is the result of an international collaborative effort carried out by a large group of institutions and members of the PRISMA Consortium. Conceptualisation: ERS, ZH, MIN, SM, NS, ASJ, BJW, VA, WKM, DG, AGC, SJB, VRM, MPK, MBS, WM, KPA, SN, CTA and LML. Funding acquisition: ERS, CM, ZH, MIN, SM, VA, AGC, SJB, MPK, MBS, WM, KPA, SN, CTA and LML. Methodology: ERS, CM, QP, JS, CRS, ZH, MIN, FJ, SM, NS, ASJ, BJW, VA, WKM, DG, AGC, SJB, VRM, MPK, MBS, WM, BF, KPA, SN and CTA. Project administration: SGB, JM, AH, NY, AK, FS, KF, NS, ASJ, JW, ZA, GO, HO, SV, JS, DJ, JA, FM, HM, BF, BV, MS, IAA, VA, EO-L and SG. Resources: CM, AH, LC, AD, MD, RV, KO, BV, HM, VA and DAG. Supervision: ERS, CM, QP, JS, CRS, ZH, MIN, FJ, AH, SM, RT, BJW, VA, WKM, DG, KO, AGC, SJB, VRM, MPK,

MBS, WM, BF, BV, KPA, SN and CTA. Writing—original draft: ERS, SGB and CRS. Writing—review and editing: ERS, CM, QP, JS, SGB, JM, CRS, ZH, MIN, FJ, AH, NY, AK, FS, KF, LC, AD, MD, RV, SM, NS, ASJ, RT, BJW, VA, WKM, JW, DG, ZA, GO, HO, KO, AGC, SJB, VRM, BV, SV, JS, DJ, JA, MPK, MBS, WM, FM, BF, BV, MS, KPA, SN, CTA, IAA, VA, DAG, EO-L, SG and LML. ERS will act as the guarantor.

**Funding** This work is supported by the Bill & Melinda Gates Foundation, grant numbers (INV-047400 to KPA, CTA and SN; INV-057219 to VA; INV-043092 to SB; INV-005776 to VPT; INV-057220 to ZH; INV-016221 to MPK; INV-057222 to WM; INV-041999 to ERS; INV-060797 to CNM; and INV-057223 to SM) and the National Institutes of Health Fogarty International Center (K01TW012426 NIH/FIC to MBS).

**Competing interests** None declared.

**Patient and public involvement** Patients and/or the public were involved in the design, or conduct, or reporting, or dissemination plans of this research. Refer to the Methods section for further details.

**Patient consent for publication** Not applicable.

**Provenance and peer review** Not commissioned; externally peer reviewed.

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
