## [Reviewer comments · BMJ Open]

ARTICLE DETAILS

Title (Provisional)

Protocol for the Pregnancy Risk, Infant Surveillance, and Measurement Alliance (PRISMA) Maternal and Newborn Health Study: A multi-site, prospective, open cohort study of pregnancy and postpartum health outcomes in South Asia and sub-Saharan Africa

Authors

Maternal and Newborn Health Study Investigators, PRISMA; Smith, Emily R.

VERSION 1 - REVIEW

Reviewer	1
Name	Theron, Gerhard
Affiliation	University of Stellenbosch, Obstetrics and Gynaecology, Tygerberg Hospital
Date	23-Jun-2025
COI	None

Please note: Page numbers used in the review correspond to the page numbers that appear as headers in the protocol.

General

Addressing maternal, neonatal and infant health with a prospective population-based study is most relevant and much needed research. The study design and protocol are adequate to answer the research questions. The introduction does summarize the importance of the study: “Our data will inform subsequent research to understand priority issues and to test innovative strategies that optimize pregnancy outcomes, including defining more representative and gestational-age-specific reference values for select biomarkers” (page 8, sentences 136-139).

Combining scheduled or ad hoc visits with study visits increases the feasibility of the study. “Each tool corresponds to a scheduled or ad hoc study visit conducted either at a study facility or at home and contains information on approximately 4,000 core variables” (page 12, sentences 231-233).

Shortfalls of the submitted protocol

The dates of the study should be included in the manuscript. The only dates are included when reference are made to different ethics committees that approved the protocol (2020 to 2024). The authors must include a projected timeline in the protocol.

The authors do not describe sharing of information with the resident health professionals of results collected by the study teams during the study that may impact quality of care to patients included in the study. This applies particularly to results of diagnostic tests for febrile illnesses, micronutrient deficiencies and special investigations that will be conducted as part of the study and not routinely done as part of standard of care in the regions where the study is conducted.

The protocol does mention that at least some of the special investigations will have results available while participants are pregnant. “At the 28-week antenatal visit, an oral glucose tolerance test will be done to screen for gestational diabetes among participants with normal hemoglobin A1c levels at enrollment” (page 15, sentences 271-272). The protocol must provide information of which blood specimen will be kept in a repository for later analysis and which will be done real time.

Placental function has not been included in the protocol. This can easily be done with measuring umbilical artery flow velocity wave forms at the repeat ultrasound to measure fetal biometry recommended at 28 weeks. Most ultrasound scanners do include the technology to do this and the sonographers most likely are trained to do it.

Abstract

Sentences 35 to 42: “Robust clinical data will be collected at up to 12 study visits during antenatal care, labor and delivery, and through one year postpartum. A total of 34 outcomes will be captured.”

Methods and analysis

Page 8, sentences 154-155: “Participants will be assessed at <20, 20, 28, 32, and 36 weeks gestation, “.

The protocol does not mention the 12 visits limitation as mentioned in the abstract. Patients with pregnancy complication may exceed 12 antenatal visits and there is no reason to only allow data collection for 12 visits.

Page 9, sentences 158-159: “Verbal autopsies will be conducted in the case of maternal, fetal, neonatal, or infant death to determine cause.” The causes of death must be determined by scrutiny of clinical notes as well as information gained from verbal autopsies and not only verbal autopsies.

Table 1

Include severe maternal morbidity (maternal near miss) as an outcome criterium. Severe maternal morbidity data will be needed to determine composite severe maternal outcome as mentioned in the table.

VERSION 1 - AUTHOR RESPONSE

General

Addressing maternal, neonatal and infant health with a prospective population-based study is most relevant and much needed research. The study design and protocol are adequate to answer the research questions. The introduction does summarize the importance of the study: “Our data will inform subsequent research to understand priority issues and to test innovative strategies that optimize pregnancy outcomes, including defining more representative and gestational-age-specific reference values for select biomarkers” (page 8, sentences 136-139).

Combining scheduled or ad hoc visits with study visits increases the feasibility of the study. “Each tool corresponds to a scheduled or ad hoc study visit conducted either at a study facility or at home and contains information on approximately 4,000 core variables” (page 12, sentences 231-233).

Shortfalls of the submitted protocol

1. The dates of the study should be included in the manuscript. The only dates are included when references are made to different ethics committees that approved the protocol (2020 to 2024). The authors must include a projected timeline in the protocol.

Thank you for this suggestion. We have included the dates of study enrollment start in Lines 152-154: “Enrollment began in September 2022 in Pakistan, November 2022 in Kenya, December 2022 in Ghana and Kenya, June 2023 in South India, and December 2023 in North India.” As this cohort is open and ongoing, no projected enrollment dates are specified.

2. The authors do not describe sharing of information with the resident health professionals of results collected by the study teams during the study that may impact quality of care to patients included in the study. This applies particularly to results of diagnostic tests for febrile illnesses, micronutrient deficiencies and special investigations that will be conducted as part of the study and not routinely done as part of standard of care in the regions where the study is conducted.

This is an excellent point and a strength of this study that we will better clarify. Please see the updated text on Lines 180-185: “Protocol implementation is jointly carried out by clinical teams at participating health facilities and study staff from partnering research institutions (i.e. the Kenya Medical Research Institute, Kintampo Health Research Centre, University of North Carolina Global Projects Zambia, Christian Medical College Vellore, Society for Applied Studies, and Aga Khan University). Clinical assessments and subsequent findings conducted by study staff will be integrated with clinical care.”

3. The protocol does mention that at least some of the special investigations will have results available while participants are pregnant. “At the 28-week antenatal visit, an oral glucose tolerance test will be done to screen for gestational diabetes among participants with normal hemoglobin A1c levels at enrollment” (page 15, sentences 271-272). The protocol must provide information of which blood specimen will be kept in a repository for later analysis and which will be done real time.

We agree that it is an important distinction which study results will be provided to participants real time versus stored and batch tested later. Please see the edits in the

'Laboratory and Diagnostic Tests' section that clarify this (Lines 270-276, 277-280, and 286-289).

4. Placental function has not been included in the protocol. This can easily be done with measuring umbilical artery flow velocity wave forms at the repeat ultrasound to measure fetal biometry recommended at 28 weeks. Most ultrasound scanners do include the technology to do this and the sonographers most likely are trained to do it.

Kindly note that this protocol does not currently measure placental function, though some study sites have additional studies focused on assessing additional ultrasound-based parameters. Ultrasound exams will be conducted by sonographers to measure fetal biometry.

Abstract

5. Sentences 35 to 42: "Robust clinical data will be collected at up to 12 study visits during antenatal care, labor and delivery, and through one year postpartum. A total of 34 outcomes will be captured."

Thank you for pointing this out. We have clarified the language in Line 41: "Robust clinical data will be collected at 12 **scheduled** study visits during antenatal care, labor and delivery, and through one year postpartum."

Methods and analysis

6. Page 8, sentences 154-155: "Participants will be assessed at <20, 20, 28, 32, and 36 weeks gestation" The protocol does not mention the 12 visits limitation as mentioned in the abstract. Patients with pregnancy complications may exceed 12 antenatal visits and there is no reason to only allow data collection for 12 visits.

Thank you. We have updated the text throughout to clarify that we are referring to 12 scheduled study visits and that any additional unscheduled visits will also be documented. Please see Lines 158, 161-162, and 226.

7. Page 9, sentences 158-159: "Verbal autopsies will be conducted in the case of maternal, fetal, neonatal, or infant death to determine cause." The causes of death must be determined by scrutiny of clinical notes as well as information gained from verbal autopsies and not only verbal autopsies.

We agree that this is an important consideration. While using a physician coding approach has some value, it can also introduce a high degree of variability between study sites. For purposes of harmonization, we will use the interVA results as the primary cause of death for this study. We have updated Lines 165-166 accordingly: "Verbal autopsies will be conducted in the case of maternal, fetal, neonatal, or infant death and analyzed by InterVA computer software to ensure a harmonized approach to cause of death determination."

Table 1

8. Include severe maternal morbidity (maternal near miss) as an outcome criterion. Severe maternal morbidity data will be needed to determine composite severe maternal outcomes as mentioned in the table.

We agree and respectively point out that our composite “severe maternal outcomes” definition includes mate

VERSION 2 - REVIEW

Reviewer	1
Name	Theron, Gerhard
Affiliation	University of Stellenbosch, Obstetrics and Gynaecology, Tygerberg Hospital
Date	10-Oct-2025
COI	

The authors must be complemented with careful consideration of the points mentioned by the reviewer.

The answer to point 4 mentioned in the review do require a comment: "Kindly note that this protocol does not currently measure placental function, though some study sites have additional studies focused on assessing additional ultrasound-based parameters. Ultrasound exams will be conducted by sonographers to measure fetal biometry." Measuring fetal biometry will only partially address fetal compromise. Adding measuring placental function with umbilical artery flow velocity wave forms will result in earlier detection of fetal compromise and further improving perinatal outcome. Measuring function is more important than measuring size.

VERSION 2 - AUTHOR RESPONSE

Comments to the Author—Reviewer 1

1. The authors must be complemented with careful consideration of the points mentioned by the reviewer. The answer to point 4 mentioned in the review does require a comment: "Kindly note that this protocol does not currently measure placental function, though some study sites have additional studies focused on assessing additional ultrasound-based parameters. Ultrasound exams will be conducted by sonographers to measure fetal biometry." Measuring fetal biometry will only partially address fetal compromise. Adding measuring placental function with umbilical artery flow velocity wave forms will result in earlier detection of fetal compromise and further improving perinatal outcome. Measuring function is more important than measuring size.

Thank you for the comment. To provide additional clarification, although the primary study outcomes of interest collected during the ultrasound exams are pregnancy

viability, gestational age determination, and fetal biometry, other metrics of clinical importance are assessed at all ultrasound visits as well. These metrics include but are not limited to: fetal heart rate activity, number of fetuses, fetal lie, amniotic fluid index, suspected fetal malformation or anomaly, placental localization and previa assessment. Unfortunately, we do not currently document umbilical artery flow velocity per our current protocol, as not all study site machines have doppler capabilities. However, this could be considered for future iterations.

2. If you have selected 'Yes' above, please provide details of any competing interests:
Not applicable.

The authors have declared no conflicts of interest.